

# Influence of Lymphangio vascular (V) and perineural (N) invasion on survival of patients with resected esophageal squamous cell carcinoma (ESCC): a single-center retrospective study

Chengke Xie[1,*], Zhiyao Chen[2,*], Jie Xu[2], Zhiyong Meng[3], Zhijun Huang[2] and Jianqing Lin[4]

[1] Department of Gastrointestinal & Esophageal Surgery, The Second Affiliated Hospital of Fujian Medical University, Quanzhou, China
[2] Department of Gastrointestinal & Esophageal Surgery, The Second Affiliated Hospital of Fujian Medical University, Quanzhou, Fujian, China
[3] Department of Ophthalmology, The Second Affiliated Hospital of Fujian Medical University, Quanzhou, Fujian, China
[4] Department of Thyroid & Breast Surgery, The Second Affiliated Hospital of Fujian Medical University, Quanzhou, Fujian, China
* These authors contributed equally to this work.

Corresponding authors
Zhijun Huang, huangzj@fjmu.edu.cn
Jianqing Lin,
ljq13905977336@163.com

## ABSTRACT

**Background:** Lymphangio vascular invasion (LVI) and perineural invasion (PNI) are associated with survival following resection for gastrointestinal cancer. But the relationship between LVI/PNI and survival of esophageal squamous cell carcinoma (ESCC) is still unclear. We aim to demonstrate the prognostic significance of LVI/PNI in ESCC.

**Methods:** A total of 195 ESCC patients underwent curative surgery from 2012 to 2018 was collected in the 2nd Affiliated Hospital of Fujian Medical University. All the patients were divided into four groups based on the status of the neurovascular invasion: (1) neither LVI nor PNI (V0N0); (2) LVI alone (V1N0); (3) PNI alone (V0N1); (4) combined LVI and PNI (V1N1). First, the analysis included the Kaplan-Meier survival estimates with the Log rank test were performed to determine median overall survival (OS) in different groups divided according to the clinical factor, respectively. And the association between OS with multi clinical factors was examined using Cox regression analysis. Next, the risk factors for recurrence in patients with V1N1 were analyzed with univariate and multivariate logistic regression analyses, respectively.

**Results:** The cases in V0N0, V1N0, V0N1, and V1N1 groups were 91 (46.7%), 62 (31.8%), 9 (4.6%) and 33 (16.9%), respectively. The OS in the four groups was different ($P < 0.001$). The 1-, 3- and 5-year OS in V0N0 group was higher than that in V1N1 group, respectively (1-year OS: 93.4% *vs* 75.8%, 3-year OS: 53.8 % *vs* 24.2%, 5-year OS: 48.1% *vs* 10.5%). The OS in stage I-II for patients with V1N1 was significantly lower than that in the other groups (V0N0, V1N0, V0N1) ($P < 0.001$). The postoperative adjuvant chemotherapy was a significant impact factor of OS for ESCC patients with V1N1 ($P = 0.004$). Lymphatic invasion and LVI were

significantly prognosis factors associated ($P = 0.036$, $P = 0.030$, respectively). The ulcerative type is a risk factor for V1N1 occurance ($P = 0.040$).

**Conclusions:** The LVI and PNI are important prognosis factors for ESCC patients. ESCC patients with simultaneous lymphangio vascular and perineural invasion (V1N1) showed worse OS than patients with either lymphangio vascular or perineural invasion alone (V1N0 or V0N1) or none (V0N0). In addition, adjuvant chemotherapy may prolong the OS for ESCC patients with V1N1.

# INTRODUCTION

Esophageal cancer (EC), one of the most aggressive gastrointestinal malignancies, is the 4th most common cancer and the 4th leading cause of cancer-related mortality in China (*Chen et al., 2016*; *Malhotra et al., 2017*). The main histologic type is squamous cell carcinoma in China. Surgery has been the main curative treatment of resectable EC (*Shah et al., 2020*). However, the overall prognosis is still poor, particularly in esophageal squamous cell carcinoma ESCC: the 5-year overall survival rate was only 15% to 25% (*Pennathur et al., 2013*). The node and metastasis (TNM) classification system were widely accepted to evaluate the prognosis of ESCC worldwide according to the American Joint Committee on Cancer (AJCC) guideline (*Rice et al., 2017*). The prediction of prognosis and survival rate of ESCC is based on the TNM system at present, but it is unable to evaluate accurately especially in the early stage (*Cai & Xin, 2010*; *Rice et al., 2017*). Lymphangio vascular invasion (LVI) and perineural invasion (PNI) could often be found in the postoperative pathological diagnosis of gastrointestinal tumors (*Cao et al., 2020*; *Chen et al., 2020*; *Fang et al. 2019* ; *Marx et al., 2020*). Esophageal carcinoma cells directly infiltrate and spread through blood or lymphatic vessels and nerve invasion (*Smyth et al., 2017*). This pattern is the manifestation of invasive characteristics, which are often a precursor of tumor metastasis and disease progression (*Smyth et al., 2017*). *Zhang et al. (2018)* had reported that microscopic lymphangio vascular or/and perineural invasion may be an important prognostic factor for ESCC.

LVI which including micro-lymphatic vessel invasion (MLVI) and micro-blood vessel invasion (MBVI), is a risk factor of tumor metastasis and invasion (*Gujam et al., 2014*). Studies had found that the presence of LVI was an important factor for poor clinical prognosis patients with gastric cancer, colorectal cancer, ductal breast cancer, and non-small cell lung cancer (*Fang et al. 2019*; *Gujam et al., 2014*; *Marx et al., 2020*; *Hamanaka et al., 2015*). Similarly, LVI is also an independent risk factor of prognosis for patients with ESCC (*Wang et al., 2016*; *Zhang et al., 2017*).

PNI, which was known as perineural diffusion and neurotropism, is used to describe the process of cancer cells invading, surrounding, and passing through nerves (*Batsakis, 1985*; *Schmitd, Scanlon & D'Silva, 2018*). PNI means that the changes of nerve cells and supporting cells in the background of cancer, including the changes and migration of the

perineural matrix (*Ekin et al., 2014*). PNI could enhance the flexibility, mobility, and invasiveness of tumor cells, the injury, and regeneration of nerve cells & the interaction, chemotaxis (*Chen et al., 2019*). PNI was not only the important pathway of tumor proliferation and metastasis, but also an independent risk factor affecting the of esophageal cancer prognosis (*Kim et al., 2021*; *Zhang et al., 2021*).

Compared to the traditional prognostic factors, the LVI and PNI may be the new risk factors for patients with ESCC, especially the lymph-node negative patients (*Hsu et al., 2019*; *Kim et al., 2021*). However, many studies only considered the influence of prognosis of LVI or PNI for ESCC patients, and ignored the relationship between the them, that may easily cause confusion and affect prognosis assessment. This study retrospectively analyzed the clinicopathological data of 195 ESCC patients with LVI or/and PNI, so as to provide a theoretical basis for clinicians to explore individualized treatment plans.

## MATERIALS AND METHODS

### Patients and characteristics

A total of 195 ESCC patients who underwent radical esophagectomy with two- or three- field lymphadenectomy was collected in our center from Jan 2012 to Dec 2018. The inclusion criteria were as follows: (1) without neoadjuvant chemotherapy, radiotherapy and targeted therapy in pre-operation; (2) one primary tumor only and no distant metastasis (M1) before surgery; (3) the histologic type was squamous cell carcinoma; (4) complete data including age, gender, TNM classification system (AJCC, 7th edition), pathologic information and follow up data. The study's retrospective protocol obtained the consent of patients by telephone and filled in informed consent and had been approved by the Medical ethics committee of the Second Affiliated Hospital of Fujian Medical University (IRB number 124, 2021).

All the patients were divided into four groups according to the status of the neurovascular invasion: (1) neither LVI nor PNI (V0N0); (2) LVI alone (V1N0); (3) PNI alone (V0N1); (4) combined LVI and PNI (V1N1).

The evaluation indexes included age, gender, postoperative adjuvant chemotherapy, tumor location, tumor length, tumor gross specimen type, Stage, the depth of tumor invasion, lymphatic invasion, LVI, and PNI.

### Follow-up

All the patients underwent CT-scan and hematological exam for personal interview every 3–6 months for post-operative Year 1–2, every 6–12 months for post-operative Year 3–5, and then every year beyond post-operative Year 5. The end-point event was patient death. The data on the patients without end-point events were censored until May 2021. The data for the patients who did not complete follow-up were censored at the last follow-up date.

### Pathological examination

All samples were reviewed by a team of experienced pathologists who know nothing about the patient's data for histological diagnosis and grade. The specimens were stained with

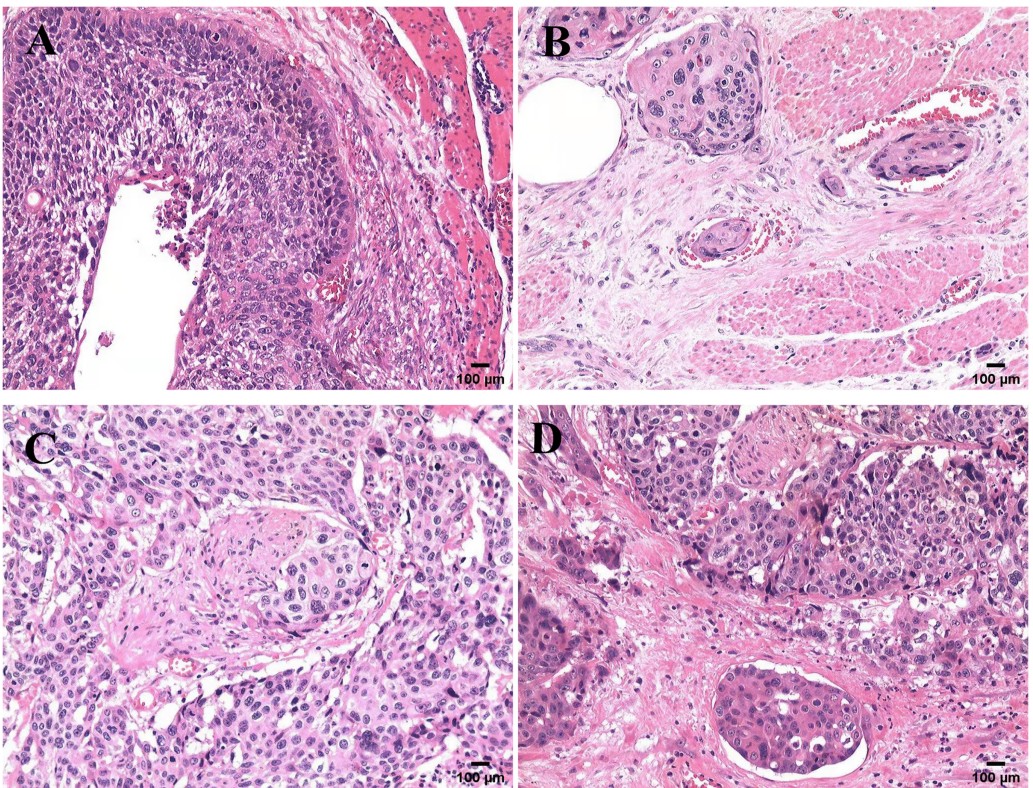

**Figure 1 Microscopic features of four pathological patterns.** Pathological section of ESCC with four histological characteristics stained with hematoxylin and eosin: (A) neither LVI nor PNI (V0N0); (B) only LVI (V1N0); (C) only PNI (V0N1); (D) combined LVI and PNI (V1N1).

hematoxylin-eosin (H & E) and examined (*Kim et al., 2021*; *Wang et al., 2016*). LVI was detected as the presence of tumor cell clusters in vessels stained positive (*Perry et al., 2015*). PNI was detected as the presence of tumor cells within any of the three layers of the nerve sheath, or close to the nerve and involving ≥ 33% of the nerve circumference (*Liebig et al., 2010*). The microscopic sections of the four histological modes are shown in Fig. 1.

## STATISTICAL ANALYSIS

Statistical analyses were applied with SPSS software version 25.0 (IBM, SPSS Statistics, Chicago, IL, USA). The differences and cumulative hazard of OS were conducted using the Kaplan-Meier method and log-rank test. The relationships of OS with characteristics (sex, age, tumor site, tumor length, tumor gross specimen type, grade, lymphatic invasion, stage, depth of tumor invasion, LVI, PNI, and postoperative adjuvant chemotherapy) were determined using univariable and multivariable Cox proportional hazards model. The optimal cut points for variables identified as influencing V1N1 were confirmed using receiver operating characteristic (ROC) curve analysis. $P < 0.05$ was considered significant.

**Table 1 Case baseline data of clinical study.**

| Clinical factors | Median | Variables | Total $n = 195$ | V0N0 $n = 91$ (46.7) | V1N0 $n = 62$ (31.8) | V0N1 $n = 9$ (4.6) | V1N1 $n = 33$ (16.9) |
|---|---|---|---|---|---|---|---|
| Age year | 60 (41–78) | <60 (%) | 92 (47.2) | 41 (21.0) | 27 (13.8) | 6 (3.1) | 18 (9.2) |
| | | ≥60 (%) | 103 (52.8) | 50 (25.6) | 35 (17.9) | 3 (1.5) | 15 (7.7) |
| Sex | | Male (%) | 140 (71.8) | 58 (29.7) | 46 (23.6) | 7 (3.6) | 29 (14.9) |
| | | Female (%) | 55 (28.2) | 33 (16.9) | 16 (8.2) | 2 (1.0) | 4 (2.1) |
| Tumor location | | Upper (%) | 28 (14.3) | 13 (6.7) | 10 (5.1) | 2 (1.0) | 3 (1.5) |
| | | Middle (%) | 97 (46.7) | 56 (28.7) | 28 (14.3) | 2 (1.0) | 11 (5.6) |
| | | Lower (%) | 70 (39.0) | 22(11.3) | 24(12.3) | 5 (2.6) | 19 (9.7) |
| Grade | | Well (%) | 36 (18.5) | 16 (8.2) | 9 (4.6) | 3 (1.5) | 8 (4.1) |
| | | Moderate (%) | 121 (62.0) | 61 (31.3) | 40 (20.5) | 5 (2.6) | 15 (7.7) |
| | | Poor (%) | 38 (19.5) | 14 (7.2) | 13 (6.7) | 1 (0.5) | 10 (5.1) |
| Depth of tumor invasion | | T1 (%) | 37 (19.0) | 27 (13.8) | 9 (4.6) | 0 | 1 (0.5) |
| | | T2 (%) | 36 (18.5) | 24 (12.3) | 10 (5.1) | 0 | 2 (1.0) |
| | | T3 (%) | 116 (59.5) | 40 (20.5) | 42 (21.5) | 8 (4.1) | 26 (13.3) |
| | | T4 (%) | 6 (3.0) | 0 | 1 (0.5) | 1 (0.5) | 4 (2.1) |
| Primary tumor length mm | 36 (8–100) | <36 (%) | 104 (53.3) | 61 (31.3) | 34 (17.4) | 3 (1.5) | 23 (11.8) |
| | | ≥36 (%) | 91 (46.7) | 30 (15.4) | 28 (14.4) | 6 (3.1) | 10 (5.1) |
| Gross specimen type | | Other (%) | 60 (30.8) | 42 (21.5) | 12 (6.2) | 2 (1.0) | 2 (1.0) |
| | | Ulcerative type (%) | 70 (35.9) | 24 (12.3) | 25 (12.8) | 2 (1.0) | 21 (10.8) |
| | | Medullary type (%) | 65 (33.3) | 25 (12.8) | 25 (12.8) | 5 (2.6) | 10 (5.1) |
| Lymphatic invasion | | N0 (%) | 91 (46.7) | 70 (35.9) | 9 (4.6) | 7 (3.6) | 5 (2.6) |
| | | N1 (%) | 60 (30.8) | 13 (6.7) | 31 (15.9) | 0 | 16 (8.2) |
| | | N2 (%) | 33 (16.9) | 7 (3.6) | 16 (8.2) | 1 (0.5) | 9 (4.6) |
| | | N3 (%) | 11 (5.6) | 1 (0.5) | 6 (3.1) | 1 (0.5) | 3 (1.5) |
| Stage | | I stage (%) | 28 (14.3) | 24 (12.) | 3 (1.5) | 0 | 1 (0.5) |
| | | II stage (%) | 69 (35.4) | 47 (24.1) | 10 (5.1) | 7 (3.6) | 5 (2.6) |
| | | III stage (%) | 83 (42.6) | 19 (9.7) | 42 (21.5) | 0 | 22 (11.3) |
| | | IV stage (%) | 15 (7.7) | 1 (0.5) | 7 (3.6) | 2 (1.0) | 5 (2.6) |
| Postoperative chemotherapy | | Yes (%) | 72 (36.9) | 23 (11.8) | 25 (12.8) | 5 (2.6) | 19 (9.7) |
| | | No (%) | 123 (63.1) | 68 (34.9) | 37 (19.0) | 4 (2.1) | 14 (7.2) |

Note:
Characteristics of patients with ESCC ($n = 195$).

# RESULTS

## Patient characteristics and prevalence of LVI and PNI

All patients included 140 (71.8%) males and 55 (28.2%) females. The median age was 60 years old. The overall prevalence of the LVI and PNI was 48.7% and 21.5 %, respectively. The patients with neither LVI nor PNI accounted for 46.7%. Patients with combined LVI and PNI accounted for 16.9%. Demographic information and the association of LVI and PNI with clinicopathological variables were showed in Table 1.

## Relationship of OS with LVI or/and PNI

As shown in Fig. 4F, the prognosis of patients with neither LVI nor PNI was superior to that with either LVI or PNI ($P < 0.001$). The average OS was $67 \pm 4.4$ months (95% CI [58.2–75.8]) for V0N0, $34 \pm 4.1$ months (95% CI [26.1–42.4]) for V1N0, $48 \pm 14.9$ months (95% CI [22.6–73.2]) for V0N1, and $32 \pm 5.1$ months (95% CI [22.3–42.5]) for V1N1.

As shown in Fig. 2, The OS in stage I–II for patients with V1N1 was significantly lower than that in the other stages (V0N0, V1N0, V0N1) ($P < 0.001$). The OS was significantly difference in four groups for patients with negative lymphatic invasion ($P = 0.005$), but the OS was no significantly difference compared between patients with V1N0 and V0N1. And the OS in stage (T1–2) for patients with V1N0 and V1N1 was lower than that in other stages ($P < 0.001$). Furthermore, the OS in stage (T3–4) for patients with V0N0 was higher than that with the other three groups (V1N0, V0N1, V1N1) ($P = 0.008$). But the OS in stage (T3–4) was no significant difference among patients with V1N0, V1N1 and, V0N1, respectively.

The 1-, 3- and 5- year OS rates of the cohort subdivided by the status of neurovascular invasion are listed in Table 2. The 1-, 3- and 5- year OS for patients with V0N0 was significant higher than that with V1N1 (OS: 93.4% *vs* 75.8%, 53.8 % *vs* 24.2%, 48.1 % *vs* 10.5 %), respectively ($P < 0.001$).

## Postoperative chemotherapy for patients with LVI or/and PNI

Of the 195 patients, 72 (36.9%) patients including 23 (11.8%) patients with V0N0, 25 (12.8) with V1N0, 5 (2.6%) with V0N1, and 19 (9.7%) with V1N1 underwent curative postoperative chemotherapy. The OS for patients who underwent curative postoperative chemotherapy with V1N1 was significantly lower than that with other three pathological groups (V1N0, V0N0, V0N1) ($P = 0.004$). But the OS was no significantly difference compared in subgroups for patients with V1N0, V0N0, and V0N1, respectively (Fig. 3).

## Univariate and multivariate analysis of survival

According to Cox regression univariate regression analysis, the results showed that the sex ($P = 0.036$), stage category ($P < 0.001$), lymphatic invasion ($P < 0.001$), PNI ($P = 0.009$), and LVI ($P < 0.001$) were closely associated with OS, as shown in Figs. 4A–4E. Multivariate analysis showed that LVI ($P = 0.030$) and lymphatic invasion ($P = 0.036$) were independent risk factors affecting the ESCC prognosis, as shown in Table 3.

## Risk factors related to patients with V1N1

The gender, primary tumor length, gross specimen types, lymphatic invasion, and stage were the predictive factors for patients with V1N1 using univariate logistic regression ($P < 0.05$). Multivariate logistic regression analysis showed that the Ulcerative type was only an independent risk factor for ESCC patients with V1N1 ($P = 0.04$) (Table 4). The AUC of the model was 0.821 (CI [0.752–0.890]) using multivariate ROC analysis (Fig. 4G).

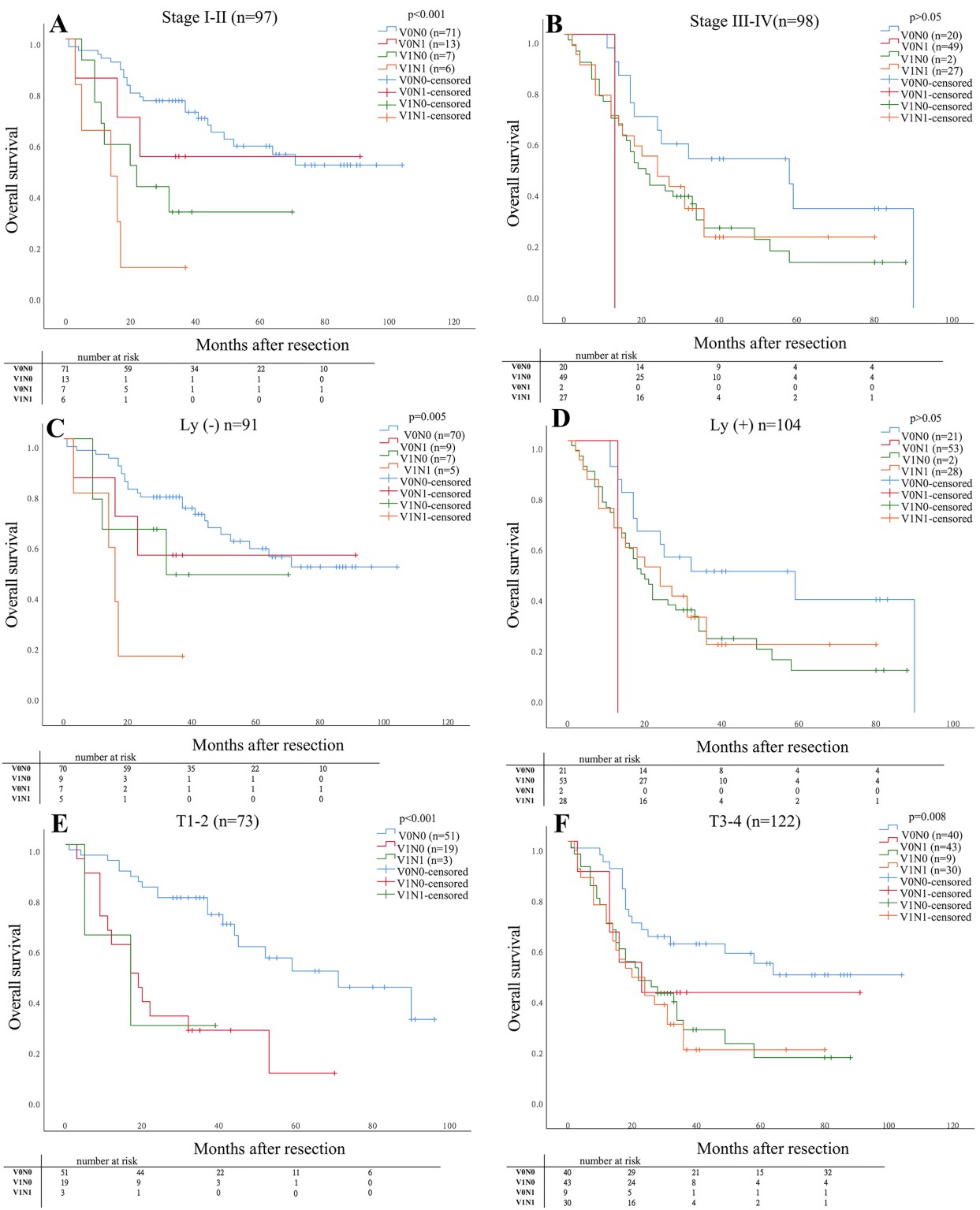

**Figure 2 The prognosis of patients was determined in specific situation compared in subgroups.** The prognosis of patients was determined in the four groups compared in subgroups (pathological stage, status of lymphatic invasion and pathological depth of tumor invasion). (A) Stage I–II $P < 0.001$; (B) Stage III–IV $P > 0.05$; (C) Ly (−) $P = 0.005$; (D) Ly (+) $P > 0.05$; (E) T1–2 $P < 0.001$; (F) T3–4 $P = 0.008$.

| Table 2 Survival rate in four pathological types. | | | | | |
|---|---|---|---|---|---|
| Time | V0N0 | V1N0 | V0N1 | V1N1 | *P*-value |
| 1-Year survival rate | 85/91 = 93.4 | 46/69 = 66.7 | 8/9 = 88.9 | 25/33 = 75.8 | 0.006 |
| 3-Year survival rate | 49/91 = 53.8 | 13/69 = 18.8 | 2/9 = 22.2 | 8/33 = 24.2 | <0.001 |
| 5-Year Survival Rate | 26/54 = 48.1 | 5/32 = 15.6 | 1/4 = 25 | 2/19 = 10.5 | 0.002 |

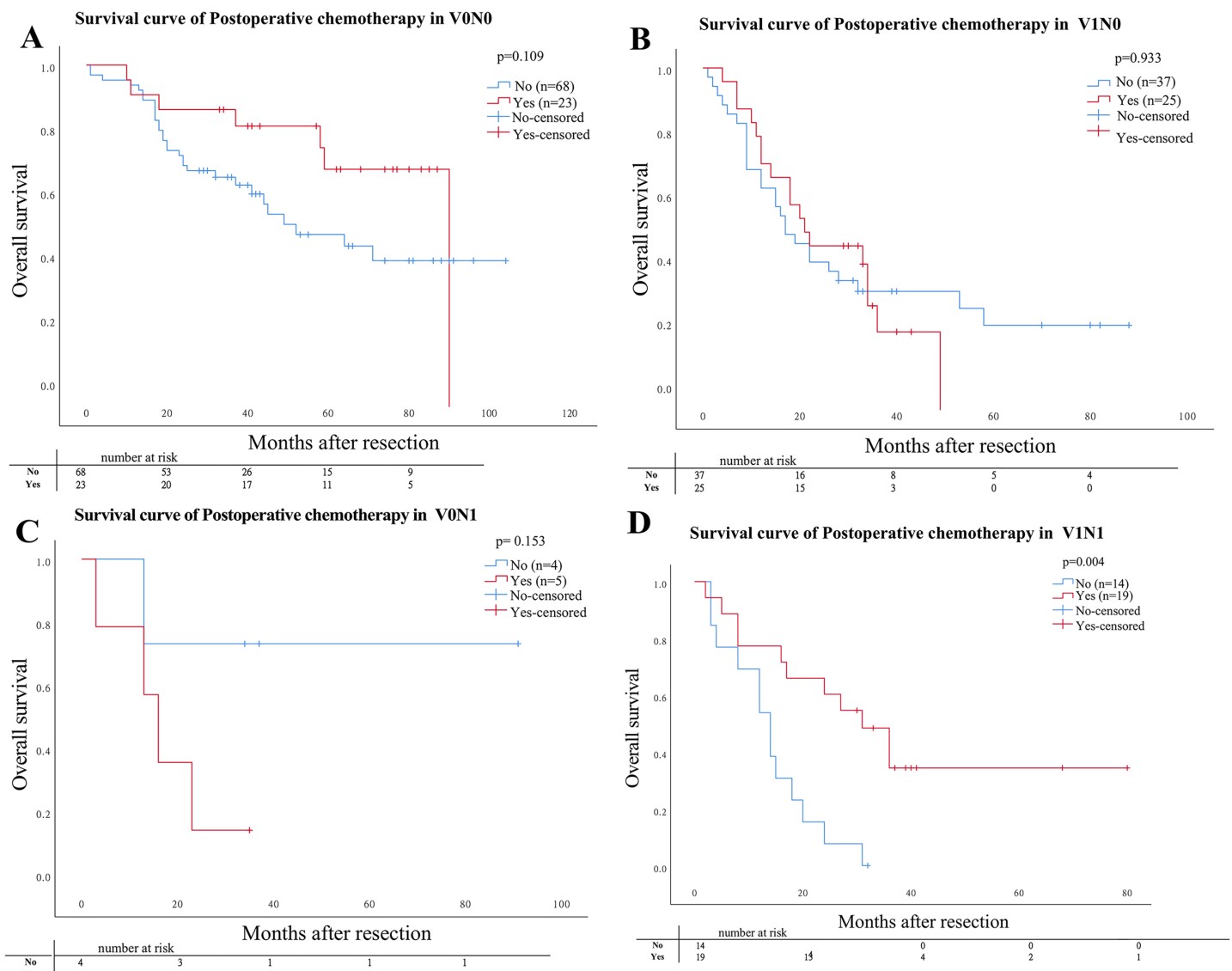

**Figure 3 Comparison of Kaplan–Meier curves stratified by whether postoperative adjuvant chemotherapy.** Comparison of Kaplan–Meier curves stratified by whether postoperative adjuvant chemotherapy. (A) V0N0 $P = 0.109$; (B) V1N0 $P = 0.933$; (C) V0N1 $P = 0.153$; (D) V1N1 $P = 0.004$.

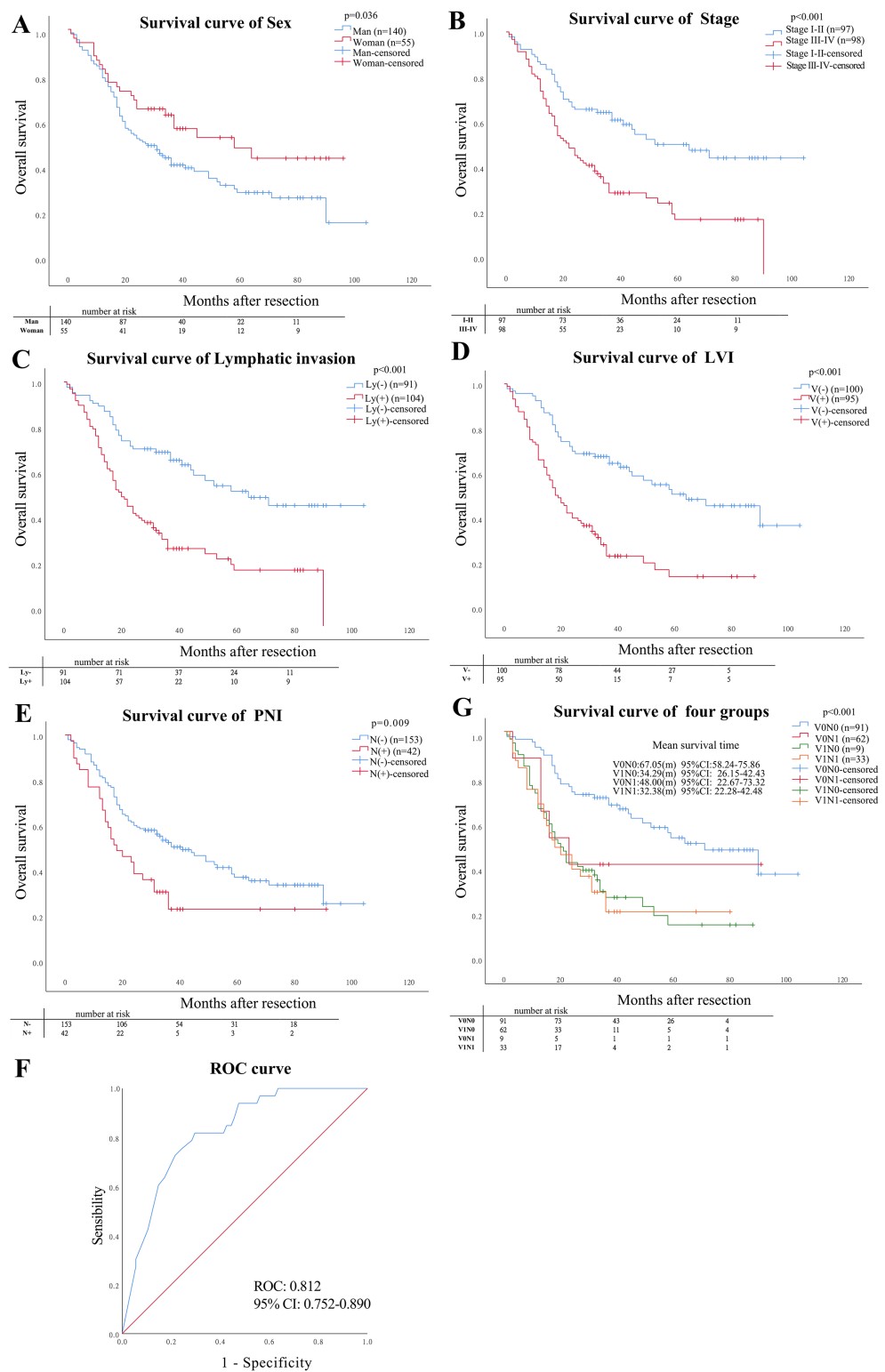

**Figure 4 Prognostic factors.** (A–E) The prognostic impact of sex, stage, lymphatic invasion status and pattern of LVI and PNI on overall survival (OS). (F) Mean survival analysis time in four pathological types, (G) the value of statistically significant variables for ESCC patients with V1N1.

**Table 3 Comparison of survival for patients.**

| Variable | Ref | Univariate analysis | | Multivariate analysis | |
|---|---|---|---|---|---|
| | | HR (95% CI) | P value | HR (95% CI) | P value |
| Age | ≥60 | 1.164 [0.799–1.697] | 0.428 | | |
| Sex | male | 0.616 [0.392–0.970] | 0.036 | 0.698 [0.437–1.114] | 0.132 |
| Tumor location | upper | | | | |
| Middle | | 0.854 [0.494–1.476] | 0.571 | | |
| Lower | | 1.079 [0.610–1.909] | 0.794 | | |
| Grade | poor | | | | |
| well | | 1.068 [0.595–1.916] | 0.826 | | |
| moderate | | 0.863 [0.532–1.400] | 0.551 | | |
| Pathological depth of tumor invasion | T1–2 | | | | |
| T3–4 | | 1.469 [0.983–2.197] | 0.061 | | |
| Stage | stage I–II | | | | |
| stage III–IV | | 2.169 [1.470–3.201] | 0.000 | 0.658 [0.301–1.437] | 0.293 |
| Tumor length (mm) | ≥36 | 1.287 [0.885–1.871] | 0.186 | | |
| Gross specimen type | Other type | | | | |
| Ulcerative type | | 1.513 [0.953–2.403] | 0.079 | | |
| Medullary type | | 1.101 [0.676–1.794] | 0.698 | | |
| Lymphatic invasion | positive | 2.551 [1.706–3.813] | 0.000 | 2.461 [1.063–5.701] | 0.036 |
| PNI | positive | 1.768 [1.153–2.711] | 0.009 | 1.159 [0.730–1.838] | 0.532 |
| LVI | positive | 2.696 [1.819–3.996] | 0.000 | 1.796 [1.057–3.053] | 0.030 |
| Postoperative chemotherapy | No | 0.924 [0.626–1.364] | 0.692 | | |

**Note:**
Compared of OS for ESCC patients based on univariate and multivariate analysis in four pathological types.

## DISCUSSION

Some studies had reported that the LVI is not only negative prognostic factor independent of tumor stage but also a high-risk factor of recurrence in ESCC patients (*Hsu et al., 2019*; *Schoppmann et al., 2013*; *Semenkovich et al., 2020*; *Wang et al., 2016*). The PNI was an adverse risk factor for survival in surgically treated patients with ESCC and reflected a more malignant behavior (*Chen et al., 2014*; *Gao et al., 2016*). Our results also showed that the LVI and PNI could be predicted factors for ESCC patients who underwent curative surgery. Our study has found that the presence of LVI and PNI were an important sign for poor clinical prognosis in lymph node negative patients. The results were the same as that in other studies (*Chen et al., 2014*; *Gao et al., 2016*; *Huang et al., 2016*; *Ye et al., 2021*). In particular, the OS in stage I-II for patients with V1N1 was significantly lower than those in the other groups (V1N0, V0N0, V0N1) (*P* < 0.001).

Based on the 2019 edition of National Comprehensive Cancer Network (NCCN) guidelines, the LVI and/or PNI are only regarded as a risk of disease recurrence (*Ajani et al., 2019*). The postoperative adjuvant therapy was not recommended for the therapy for ESCC patients with LVI and/or PNI (*Ajani et al., 2019*). But the PNI is considered as an indicator of adjuvant radiotherapy for patients with head and neck tumors (*Bakst et al., 2018*). Some studies reported that PNI could be used as one of the reference indicators of adjuvant therapy for post-operative patients with esophageal cancer (*Tsai et al., 2017*; *Gao et al., 2016*). Some scholars also proposed that the LVI is an independently significant

**Table 4 The risk factors associated with V1N1.**

| Clinical factors | Variables | V1N1 | No V1N1 | P | $\chi^2$ | Univariate analysis | | | Multivariate analysis | | |
|---|---|---|---|---|---|---|---|---|---|---|---|
| | | | | | | P | OR | 95% CI | P | OR | 95% CI |
| Age (years) | <60 | 18 | 74 | 0.000 | 22.202 | | | | | | |
| | ≥60 | 15 | 88 | | | 0.354 | 0.701 | [0.330–1.486] | | | |
| Sex | Male | 29 | 111 | 0.024 | 5.075 | | | | | | |
| | Female | 4 | 51 | | | 0.032 | 0.300 | [0.100–0.899] | 0.078 | 0.346 | [0.106–1.125] |
| Tumor location | Upper | 3 | 25 | 0.017 | 5.807 | | | | | | |
| | Middle | 11 | 86 | | | 0.926 | 1.066 | [0.276–4.120] | | | |
| | Lower | 19 | 51 | | | 0.090 | 3.105 | [0.839–11.487] | | | |
| Grade | Well | 8 | 28 | 0.088 | 4.867 | | | | | | |
| | Moderate | 15 | 106 | | | | | | | | |
| | Poor | 10 | 28 | | | | | | | | |
| Depth of tumor invasion | T1–2 | 3 | 70 | 0.000 | 13.626 | | | | | | |
| | T3–4 | 30 | 92 | | | 0.001 | 7.609 | [2.231–25.949] | 0.058 | 4.565 | [0.950–21.930] |
| Primary tumor length (mm) | <36 | 10 | 94 | 0.004 | 8.465 | | | | | | |
| | ≥36 | 23 | 68 | | | 0.005 | 3.179 | [1.421–7.114] | 0.133 | 1.968 | [0.814–4.762] |
| Gross specimen type | Other | 3 | 57 | 0.000 | 29.995 | | | | | | |
| | Ulcerative type | 20 | 50 | | | 0.002 | 7.600 | [2.131–27.104] | 0.040 | 4.302 | [1.068–17.332] |
| | Medullary type | 10 | 55 | | | 0.070 | 3.455 | [0.902–13.224] | 0.301 | 2.172 | [0.499–9.462] |
| Lymphatic invasion | Negative | 5 | 86 | 0.000 | 15.851 | | | | | | |
| | Positive | 28 | 76 | | | 0.000 | 6.337 | [2.330–17.231] | 0.052 | 10.960 | [0.977–122.917] |
| Stage | I–II stage | 6 | 91 | 0.000 | 15.828 | | | | | | |
| | III–IV stage | 27 | 71 | | | 0.000 | 5.768 | [2.259–14.728] | 0.399 | 0.357 | [0.033–3.914] |

**Note:**
The risk factors associated with cancer specific survival using Chi square test, univariate and multivariate logistic regression analysis.

prognostic factor, and can be a potential determining factor for the use of adjuvant therapy for ESCC patients who underwent esophagogastrectomy (*Schoppmann et al., 2013*; *Zafirellis et al., 2002*). Interestingly, our results showed that postoperative adjuvant chemotherapy could prolong the overall survival of ESCC patients with simultaneous LVI and PNI (V1N1). Therefore, both LVI and PNI (V1N1) maybe an indicator for adjuvant therapy for postoperative ESCC patients. However, the prospective, multicenter, randomized controlled studies still need to be carried out to verify this conclusion.

The results showed that the sex, stage, lymphatic invasion, LVI and, PNI is the prognostic factor for ESCC patients, respectively ($P < 0.05$). In addition, we found that the Lymphatic invasion and LVI were independent the prognostic factors for ESCC patients ($P = 0.036$, 0.030, respectively). Therefore, we considered that the prognosis for postoperative ESCC patients could be more predicted accurately based on the TNM system and the status of the neurovascular invasion.

The logistic regression analysis results showed that the sex, depth of tumor invasion, primary tumor length, gross specimen type, lymphatic invasion, and stage were all risk factors ($P < 0.05$) for ESCC patients with V1N1. But further analysis results showed that only the Ulcerative type is an independent risk factor ($P = 0.04$). *Lee et al. (2016)* study

reported that ulcerative EGC exhibit more aggressive behaviors than non-ulcerative EGC concerning for to submucosal invasion, LNM, LVI, and PNI. A deep ulcer more than the thickness of the adjacent mucosal surface is an important factor for the invasion of submucosal carcinoma (*Lee et al., 2012*). Therefore, the ulcerative type may be a risk factor for the occurrence of both LVI and PNI.

Different for other studies, we have made a more detailed classification of the types of the neurovascular invasion, which is helpful to evaluate the prognosis more accurately. However, our research was a single-center retrospective study with the limited number of patients excluded that received neoadjuvant therapy. This is maybe a limitation in our research.

In conclusion, the OS and prognosis of ESCC patients with four pathological characteristics are different from each other, especially the patients in Stage I–II. The OS for patients with V1N1 is the worst compared with the other groups (V1N0, V0N1 and V1N1). In addition, postoperative adjuvant chemotherapy could prolong the OS for ESCC patients with V1N1. Therefore, postoperative adjuvant chemotherapy could be recommended for the patients with V1N1. Besides, the ulcerative type may be a risk factor for the occurrence of both LVI and PNI. The LVI and PNI are important prognosis factors for ESCC patients.

### Funding

This work was supported by the Natural Science Research Foundation of Fujian Province: (No. 2019J01476 and No. 2021J01273), Medical Innovation Foundation of Fujian Province (No. 2019-CXB-19), High-level Talent Foundation of Quanzhou City (No. 2020C012R) and High-level Talent Innovation Foundation of Quanzhou City (No. 2017Z008). The funders had no role in study design, data collection and analysis, decision to publish, or preparation of the manuscript.

### Grant Disclosures

The following grant information was disclosed by the authors:
Natural Science Research Foundation of Fujian Province: 2019J01476 and 2021J01273.
Medical Innovation Foundation of Fujian Province: 2019-CXB-19.
High-level Talent Foundation of Quanzhou City: 2020C012R.
High-level Talent Innovation Foundation of Quanzhou City: 2017Z008.

### Competing Interests

The authors declare that they have no competing interests.

### Author Contributions

- Chengke Xie performed the experiments, analyzed the data, prepared figures and/or tables, and approved the final draft.
- Zhiyao Chen analyzed the data, prepared figures and/or tables, and approved the final draft.

- Jie Xu performed the experiments, prepared figures and/or tables, and approved the final draft.
- Zhiyong Meng analyzed the data, authored or reviewed drafts of the paper, and approved the final draft.
- Zhijun Huang conceived and designed the experiments, authored or reviewed drafts of the paper, and approved the final draft.
- Jianqing Lin conceived and designed the experiments, authored or reviewed drafts of the paper, and approved the final draft.

## Human Ethics

The following information was supplied relating to ethical approvals (*i.e.*, approving body and any reference numbers):

Ethics Committee of the Second Clinical Medical College Affiliated to Fujian Medical University.

## Data Availability

The raw data is available in the Supplemental File.

## Supplemental Information

Supplemental information for this article can be found online at http://dx.doi.org/10.7717/peerj.12974#supplemental-information.

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
