# Peer review of "Influence of Lymphangio vascular (V) and perineural (N) invasion on survival of patients with resected esophageal squamous cell carcinoma (ESCC): a single-center retrospective study"

_PeerJ, doi:10.7717/peerj.12974_

## Round 0.1 · original submission · Major Revisions

There are major and minor issues with this paper that need to be addressed, after which the manuscript shall be re-assessed.

As pointed out by two reviewers, there are numerous grammatical and syntax errors throughout the text. The manuscript requires revision by someone with full professional proficiency in English or an editing service to improve its clarity and readability.

The language is incomprehensible in many sections, making it difficult to follow. Please note that due to this issue, some items and mistakes may have been missed in this round and may come up in the next review round. In order to prevent multiple rounds of review, please carefully revise the paper and especially make significant improvements to its English.

Please respond to all reviewers’ comments and revise the paper accordingly. Additionally, the following points should also be clarified/revised.
- Line 92: please be specific when stating inclusion/exclusion criteria: do not use terms like “so on”; what else was included?
Line 100: to avoid confusion, do not separate clinical and histological indices and state them simply as ‘evaluated factors in this study’ or any other similar terms. An example of something that might cause confusion is including tumor location in histologic indices.
Line 100: explain “histologic type”. Based on line 91, only squamous cell carcinomas were included in the study.
In general, pay attention to the difference between the terms: “histologic” and “pathologic”.

·

Basic reporting

The article is well written. It has scientific value. We can accept the article.

Experimental design

The article is well written. It has scientific value. We can accept the article.

Validity of the findings

The article is well written. It has scientific value. We can accept the article.

Additional comments

The article is well written. It has scientific value. We can accept the article.

Reviewer 2 ·

Basic reporting

The authors analyzed and demonstrated the relation between LVI/PNI and the survival rate of ESCC. However, the writing quality for this manuscript is not acceptable for this journal.

There are many grammar mistakes including the misusage of punctuation and unclear statements. This main manuscript including the figure legends needs to be carefully edited with assistance from a scientific writer with appropriate scientific English writing skills.

Experimental design

The experimental design is well proposed with well-described statistic analysis.

Validity of the findings

There is sufficient data to support the conclusion in this manuscript.

Additional comments

Please see below for some minor comments:

1. The authors need to present the full name of a term in the first instance and follow it immediately by the abbreviation in parenthesis. For instance, the authors need to use the full name for the median OS on line 35.
2. The authors need to add scale bars in Figure 1.
3. Do not use abbreviations in the title.
4. The font size of the references doesn’t match with the rest of the manuscript, please enlarge them.
5. The authors need to use consistent formatting in the main manuscript when citing other people’s work.

·

Basic reporting

Line 62: Formatting inconsistencies: some spaces between text and parenthesis, others don't have spaces (e.g Line 62)

Line 79: recommend rephrasing this sentence

Line 134: Figure1 is not showing prognosis, it should be Figure2

Line 137: If the authors can reformat the figures and put the number of patients in each subgroup and the number at each sensor time point below the figure, that would greatly improve readability and clarity for the data and analysis

Line145: It will be a lot easier if the figures2 are labeled with panel numbers such as A, B, C, D (like Figure3 and 4), and in the main text, each result refers to a specific panel. Also in the legend, have a short description for each panel.

Line153, Line 156: grammar issue, cross out "were"

Line 157: in Figure3 legend, I recommend for authors be more precise on the subtitles for each panel instead of just putting down a P value.

Line 169: it was stated some studies reported that the LVI is not only a negative prognostic factor independent of tumor stage but there is only reference. Recommend adding additional references where appropriate.

Line 177: missing references for other studies when authors stated their results were the same as that in other studies.

Line 205: Please clarify what this sentence meant.

Experimental design

Line 159-161: This part needs more details and clarity. First of all, the title says Univariate and Multivariate Analysis of Survival, but is the P values reported here based on univariate analysis or Multivariate Analysis? Second, based on the table3 of the univariate analysis, there are other variables that also show a significant p value, such as invasion but are not reported in the main text? Does the invasion here indicate both V1N1? Please clarify. Third, what the multivariate analysis shows is not reported in the main text of the result section.

Line 165: the results need some clarity. Did the authors only look at V1N1 patients or all patients including three other subtypes?

Validity of the findings

Line 154-157: the interpretation and results do not match. The result shows only in V1N1, patients who underwent chemo had better OS. There were no direct comparisons between the histological types. I recommend authors re-write these parts to match results.

Line 214: The conclusion that post-operative adjuvant chemotherapy should 214 be done for the patients with V1N1 is too strong and not justified without proper discussion and weighing the benefit and risks involved.

Additional comments

I found many grammar mistakes throughout the manuscript and recommend English proof-editing.

---

## Round 0.2 · accepted · Accept

The issues raised in the previous round of review have been addressed and I am pleased to accept this manuscript.

Reviewer 2 ·

Basic reporting

The authors have addressed my concerns with the original manuscript.

Experimental design

The authors have addressed my concerns with the original manuscript.

Validity of the findings

The authors have addressed my concerns with the original manuscript.

Additional comments

I just have some minor comments about the formatting:

1. For reference (line 343-345), the format should be "Hamanaka R, Yokose T, Sakuma Y, Tsuboi M, Ito H, Nakayama H, Yamada K, Masuda R, Iwazaki M. Prognostic impact of vascular invasion and standardization of its evaluation in stage I non-small cell lung cancer. Diagn Pathol. 2015 Apr 2;10:17. doi: 10.1186/s13000-015-0249-5. PMID: 25884820; PMCID: PMC4413537." instead of "Rurika, Hamanaka, Tomoyuki, Yokose, Yuji, Sakuma, Masahiro, Tsuboi, Hiroyuki, and Pathology IJD. 2015. Prognostic impact of vascular invasion and standardization of its evaluation in stage I non-small cell lung cancer. Diagn Pathol 10 1-10 DOI 10.1186/s13000-015-0249-5."

2. Please add spaces between text and parenthesis, for example, line 59, 61, 63, 65, 67, 69, 71, 73, 75 in the first paragraph of Introduction section.

·

Basic reporting

Much improved and addressed all my previous comments and suggestions. Therefore no more comments.

Experimental design

Much improved and addressed all my previous comments and suggestions. Therefore no more comments.

Validity of the findings

Much improved and addressed all my previous comments and suggestions. Therefore no more comments.